# Mechanisms of Environmental Regulation's Impact on Green Technological Progress—Evidence from China's Manufacturing Sector

**Weijiang Liu [1], Mingze Du [1,*] and Yuxin Bai [2]**

[1] Center for Quantitative Economics of Jilin University, Changchun 130012, China; liuwj@jlu.edu.cn
[2] School of Environment and Natural Resources, Renmin University of China, Beijing 100872, China; byxin@ruc.edu.cn
* Correspondence: dennisdu520@163.com; Tel.: +86-18644988528

**Abstract:** As the world's largest developing country, and as the home to many of the world's factories, China plays a crucial role in the sustainable development of the world economy regarding environmental protection, energy conservation, and emission reduction issues. Based on the data from 2003–2015, this paper examined the green total factor productivity and the technological progress in the Chinese manufacturing industry. A slack-based measure (SBM) Malmquist productivity index was used to measure the bias of technological change (BTC), input-biased technological change (IBTC), and output-biased technological change (OBTC) by decomposing the technological progress. It also investigated the mechanism of environmental regulation, property right structure, enterprise-scale, energy consumption structure, and other factors on China's technological progress bias. The empirical results showed the following: (1) there was a bias of technological progress in the Chinese manufacturing industry during the research period; (2) although China's manufacturing industry's output tended to become greener, it was still characterized by a preference for overall $CO_2$ output; and (3) the impact of environmental regulations on the Chinese manufacturing industry's technological progress had a significant threshold effect. The flexible control of environmental regulatory strength will benefit the Chinese manufacturing industry's technological development. (4) R&D investment, export delivery value, and structure of energy consumption significantly contributed to promoting technological progress. This study provides further insight into the sustainable development of China's manufacturing sector to promote green-biased technological progress and to achieve the dual goal of environmental protection and healthy economic growth.

**Keywords:** China's manufacturing sector; green total factor productivity; biased-technical progress; environmental regulation

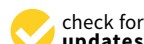



## 1. Introduction

As a pillar industry of the economy, the manufacturing industry has laid a good foundation for China's vigorous economic development and has earned a reputation as the world's factory. In the past 20 years, China's economy has grown rapidly. In 2019, the manufacturing industry's added value reached 26.9 trillion yuan (Data resources come from the World Bank database: https://data.worldbank.org.cn/ (accessed on 26 January 2021)), but behind the rapid growth are high energy consumption, high emissions, and high investment. In 2007, China replaced the United States as the world's largest $CO_2$ emitting country. The manufacturing industry's energy consumption and carbon emissions account for one-third of China's total energy consumption and carbon emissions and two-thirds of the entire industrial sector. In recent years, developed countries have been transferring low-end manufacturing and labor-intensive manufacturing to developing countries such as China through foreign investment. At the same time, they have formulated a series of environmental constraints, green barriers, and other policies to restrict. The contradiction

between environment and resource, as well as international constraints, has forced China to face severe pressure of economic transformation. To achieve sustainable economic development and contribute to global environmental problems, it is necessary to embark on a green and sustainable economic development mode. As an important part of China's economy, the manufacturing industry has become the key area of China's green economic development. The root of the change in development mode is to improve green total factor productivity. Therefore, the right way to realize China's green development is to promote the green TFP of China's manufacturing industry through reasonable environmental regulation. This paper analyzes the influence mechanism of environmental regulations on technological progress toward greening so as to achieve the dual goals of environmental protection and healthy economic development.

Environmental regulations are the driving force for China's green development and have been proven to promote green total factor productivity in China's manufacturing industry [1–4]. Neoclassical economic growth theory suggests that technological progress can promote economic growth. However, technological progress under this theory is assumed to be exogenous and neutral. In fact, different preferences for input factors due to changes in technology or differences in factor endowments lead to several biases in technological progress during the actual production. For example, Acemoglu [5] defines the technological progress bias as changing the marginal rate of substitution (MRS) among inputs, thereby increasing the marginal output of factors and achieving the optimal allocation of scarce resources. Many studies have shown that technological progress in many regions or industries is non-neutral [6–9], Kaneko [10], Zhao [11], Fujii [12], have studied the relationship between technological progress bias in China. However, these studies have not paid much attention to the impact of the direction of the technological bias on the environment and resources. In actual economic operation, when technological progress tends to input fewer resources and produce fewer undesirable outputs (pollutants, etc.), it can effectively promote economic growth while achieving energy conservation and emission reduction.

Compared with previous studies, this paper has three significant contributions. Firstly, it examines the green TFP, technological progress, and corresponding technological progress biases, including input and output technological progress biases in China's manufacturing subsectors from 2003 to 2015 based on SBM and Malmquist methods. The focus on input and output bias avoids the one-sidedness of the technology bias research [13], while the study of expected and unexpected output is more in line with the requirements for China's sustainable economic development. Secondly, based on summarizing the green TFP growth and technology bias in China's manufacturing industry, we focus on combining input and output factors along with technology bias. The trends of technology bias and the factors of input and output during the research period were analyzed to better understand the sources of technological progress and the current development status of this sector. Finally, different from the existing studies, the influence mechanism of environmental regulations and other factors such as the proportion of state-owned assets, R&D investment, and export delivery value, etc., on technological progress and technological bias help us gain a deeper understanding of the flexible control of the intensity of environmental regulations and the influence mechanism of other influencing factors on technological progress under environmental regulations, to have detailed theoretical support for further rational policymaking.

The paper is organized as follows: Section 2 provides a brief review of existing research; Section 3 introduces the main methods and data involved in this paper, including the SBM model dealing with undesirable output, the Malmquist productivity index, and the theory of technological progress bias, including the decomposition process of technological bias and the identification of technological bias. Section 4 summarizes relevant research findings on the Chinese manufacturing industry and analyzes the mechanisms underlying environmental regulations' influence on technological progress. Section 5 presents relevant conclusions and policy recommendations based on this study.

## 2. Literature Review

### 2.1. Related Research on the Environmental Regulation and Green Total Factor Productivity

Environmental regulation occurs when, given the external diseconomy of environmental pollution, the government leads relevant policies or measures to regulate the production behaviors of enterprises. By means of making enterprise production decisions incorporate resource and environmental factors into their cost considerations, achieving the goal of maintaining the coordination between environmental and economic development [14]. As environmental regulation affects enterprise behaviors directly, compared to total factor productivity (TFP), the green total factor productivity obtained by incorporating environmental pollution and energy consumption into the analytical framework of total factor productivity can represent the production level of enterprises considering environmental factors more [15–17]. With the increasingly severe environmental problems, many scholars have explored the relationship between environmental regulation and green total factor productivity. For example, Ghosal et al. [18] use data from the Swedish paper industry to research the impact of environmental regulations and enforcement policies on green total factor productivity (TFP) growth and its components related to efficiency change and technical change. The results show that environmental regulation is beneficial to the increase and sustainable production practices of green total factor productivity. Via sorting out these studies, it can be broadly divided into two categories. One is based on the type of environmental regulation. Xie et al. [19] analyzed the impacts of different types of environmental regulations on local green TFP in China, supporting the "strong" Porter Hypothesis that reasonable stringency of environmental regulations may enhance rather than lower industrial competitiveness. Cai and Zhou [20] tested the direct and indirect effects of three kinds of environmental regulations on green TFP, meanwhile testing the "Porter hypothesis". According to the impact of different types of environmental regulations on the threshold value of total factor productivity, Lei and Wu [21] found that the effects of different kinds of environmental regulations on the behavior and degree near the threshold were inconsistent. The other is based on the perspective of environmental regulation strength. For example, Zhang et al. [22] measured the total factor productivity of China's industrial sectors and tested the relationship between environmental regulation strength and total factor productivity. They found a co-integration relationship between environmental regulation and TFP, and environmental regulation's positive promotion to TFP is much more apparent in the long run than that in the short run. Li and Tao [23] divided China's 28 manufacturing sectors into three categories: heavily polluting industries, moderately polluting industries, and lightly polluting industries. Based on measuring the environmental regulation strength and green total factor productivity of three industrial sectors from 1999 to 2009, the panel data model was used to test the relationship between environmental regulation and green total factor productivity, exploring China's optimal environmental regulation strength manufacturing industry. Through the study of Chinese industrial green total factor productivity and environmental regulation strength, Li et al. [24] found environmental regulation strength has an obvious "threshold effect". When the environmental regulatory strength is lower than the "threshold value", it has no significant impact on green TFP, while when it is higher than a certain "threshold value", it will have adverse effects. Only the environmental regulatory strength between the two thresholds is conducive to the growth of green TFP. Some researchers also measured the environmental regulations from the perspective of pollution removal rate and applied them to their impact on TFP. For example, Li and Wu [25] analyzed the effect of green total factor productivity based on different urban attributes and civil environmental laws and regulations, indicating that the government should reduce market intervention, promote enterprises' technological innovation, and put forward targeted emission reduction policies.

From the review of the previous literature noted above, most of the researchers study the effects of environmental regulation on green TFP directly but seldom analyze the impact mechanism of environmental regulation, especially from the perspective of technological progress and its decomposition items. According to existing studies, technological progress

is an important part of total factor productivity, as well as the main driving force of total factor productivity growth [1,5,6], while environmental regulations will increase the cost of enterprises [26], resulting in the extrusion effect of enterprises' R&D and other expenditures, which influence the technological progress. Therefore, it is necessary to explore the influence mechanism of environmental regulations on technological progress, especially on the decomposition items of technological progress. Decomposing technological progress into biased technological progress and scale technological progress, and analyzing the decomposed items can clarify the internal impact of environmental regulations on technological progress effectively.

### 2.2. Related Research on the Bias of Technological Change

According to Fare et al. [27], we can decompose technological progress into technological scale changes and technological progress bias. Technological progress bias has been a hot topic in TFP research in recent years. If the assumption of neutrality in technological progress is not met, the possibility of bias in both input and output technologies must be considered. Many scholars have empirically studied the performance of technology bias in various fields. Sato and Morita's [28] research on labor-saving technological innovations in Japan and the United States found that such advances were beneficial to productivity. Yang et al. [29] used the stochastic frontier approach (SFA) to study the technology bias and elasticity of substitution of R&D activities in the Chinese manufacturing industry. Yoshida et al. [30] studied the productivity and technology bias of Japanese airports. They showed that the traditional growth accounting method of neutral change is not suitable for analyzing the productivity changes at Japanese airports. However, none of these studies took into account environmental and energy factors. As environmental problems have worsened in recent years, incorporating environmental and energy factors into the production framework to analyze technology bias has become an area of interest for many researchers. For example, Briec et al. [31] examined the property of productivity growth and technological bias of hydroelectric plants in Portugal, Li et al. [32] studied the green total factor productivity and technological bias of industrial water resources in China.

Based on a summary of relevant studies, the most popular methods for measuring technology bias are Data Envelopment Analysis (DEA), Stochastic Frontier (SFA), and Standardized Supply-Side System (CES). Compared to CES and SFA, DEA can avoid the bias caused by predetermined production model settings and is more accurate in measuring technology bias because it applies to multiple inputs and outputs, making it a popular method.

### 2.3. Related Research on Manufacturing Sector Sustainability Measurement

In recent years, the issue of sustainability in manufacturing has not received much attention from researchers. Most of them have focused on the impact of tariffs, trade, or resource allocation on manufacturing productivity [33–35]. Alternatively, the role of R&D spillover effects on productivity has been analyzed from the R&D perspective [36]. Among the studies on the relationship between environment and productivity in the manufacturing sector, some scholars have studied from the perspectives of environmental regulations, green innovation, and R&D, such as Marchi [37] found the important influence of technology R &D cooperation on environmental innovation tendency in his research on Spanish manufacturing industry. Brunnermeier [38] analyzed the determinants of environmental innovation in the US manufacturing industry, such as pollution expenditures and environmental regulations, and found that environmental innovation is more likely to occur in internationally competitive industries. Rubashkina [39] tested the Porter hypothesis for the European manufacturing industry by showing that environmental regulations positively impact green innovation.

For the Chinese manufacturing industry, in addition to Cao [16] and Shi [17], Cao et al. [40] studied the green growth and environmental regulations of Chinese manufacturing and found that the impact of environmental regulations on green growth is U-shaped

nonlinear. The impact in various industries is heterogeneous. Cheng et al. [41] divided the subsectors of China's manufacturing industry according to the pollution intensity and studied the green energy-environmental efficiency of the subsectors under different pollution intensities. They found that the most efficient is the lightly polluting industry, and the research, development, and governance transformation investment can promote the growth of TFP. Gong et al. [42] systematically reviewed the environmental regulation and the theoretical mechanism of the comparative advantage of trade in the green transformation and upgrading of China's manufacturing industry and proved that under the constraints of environmental regulation, the comparative advantage of trade would promote the green transformation of the manufacturing industry. Meng et al. [43] analyzed the role of environmental regulation and green innovation in the intelligent development of China's manufacturing industry, and the results showed that environmental regulation and green innovation have a positive effect on the intelligent upgrading of manufacturing.

Research on technology bias in manufacturing has mostly focused on the bias between labor and capital factors, such as Wang et al. [44], Manasse and Stanca [45]. Only a few researchers have examined the relationship between environmental regulatory policies on green TFP and technological progress in the manufacturing sector, as argued by Jaffe et al. [46], Yang et al. [47] found that environmental regulations are positively correlated with R&D expenditures and that green technology advancement has a positive impact on productivity through a study of the Taiwanese manufacturing industry. In the study of the relationship between environmental regulations and the impact of technology bias, Zhou et al. [48] analyzed the mechanism of the effect of environmental regulations on the technological progress of energy and environmental bias, showing that the impact of different types of environmental regulations on technological bias differs significantly. Song et al. [49] constructed a definition of technological progress related to the environment and measured China's environmental-biased technological progress. Song and Wang [50] found that environmental regulation, population aging, and other factors can promote green-oriented technological progress. What's more, Calel and Antonie [51] found that environmental policies can promote targeted technological change by examining the European market.

By reviewing the existing studies, it is found that there are relatively few studies that give full consideration among environmental regulation, green TFP, and technology advancement bias in the Chinese manufacturing industry, and there is no detailed arrangement of technology bias in this sector, including input-output technology bias. In addition, the mechanism of the intrinsic influence of environmental regulations on technological progress is still unclear, especially the influence on the technological bias. The "Porter's Hypothesis" and the environmental Kuznets curve indicate that environmental regulations do not have a simple linear effect on the economy. Therefore, it is more important for the green development of the manufacturing industry to investigate the mechanism of environmental regulations on technological progress, especially on the technological bias. After exploring the input/output factor combinations and input/output technology bias in this sector, this paper analyzes the influence mechanism of environmental regulations on technical progress and technology bias, providing rationalization suggestions for the sustainable development of the Chinese manufacturing industry.

## 3. Materials and Methods

### 3.1. The Model

To measure the bias of technological progress, Fare et al. [52] decomposed the Malmquist index into technological efficiency change, technological change, and decomposed technological change into technological scale change, input-biased technological progress, and output-biased technological progress, but this decomposition method does not apply to the production model with undesirable outputs. Therefore, this paper adopts the SBM model with undesired outputs proposed by Tone [53], which consider the relationship

between inputs, outputs, and undesirable outputs, solving the slack problem in efficiency evaluation better.

In this study, we treat each of the 27 industries in the Chinese manufacturing industry as a production decision unit (DMU) to construct the optimal time boundaries of Chinese manufacturing production for any period. Using inputs m for each industry, recorded as $X_k = (x_{1k}, x_{2k}, \ldots, x_{mk})$, we obtain p desired outputs, recorded as $Y_k = (y_{1k}, y_{2k}, \ldots, y_{pk})$, and q undesired outputs recorded as $B_k = (b_{1k}, b_{2k}, \ldots, b_{qk})$.

$$p(x) = \{(x, y, b) |: x \geq X\lambda, y \geq Y\lambda, b \geq B\lambda, \lambda \geq 0\} \tag{1}$$

According to model (1), the SBM model of undesired output is as follows:

$$\min \rho_k = \frac{1 - \frac{1}{m}\sum_{i=1}^{m} \frac{s_i^{x-}}{x_{ik}}}{1 + \frac{1}{p+q}\left(\sum_{r=1}^{p} \frac{p_r^{y+}}{y_{rk}} + \sum_{t=1}^{q} \frac{z_t^{b-}}{b_{tk}}\right)} \tag{2}$$

$$s.t. X\lambda + s^{x-} = x_{k'} Y\lambda - s^{y+} = y_{k'} B\lambda + s^{b-} = b_{k'}, \lambda \geq 0, s^{x-}, s^{y+}, s^{b-} \geq 0$$

where $s^{x-}, s^{y+}, s^{b-}$ represents the slack values of the input, good output, and bad output, respectively. $x_{mk}, y_{pk}, b_{qk}$ represents the $m_{th}$ input of the $k_{th}$ DMU, the $p_{th}$ desired output, and the $q_{th}$ undesired output. $\rho_k$ is a variable between 0 and 1 representing the efficiency of the $k_{th}$ DMU environment, less than 1 means that the $k_{th}$ DMU is inefficient.

This study constructs the Malmquist index's distance function in conjunction with the SBM model dealing with undesirable output. According to the Malmquist exponential decomposition method of Fare, [8] the TFP growth rate is decomposed into technological change, efficiency change, further, decomposed into output-biased technological progress and input-biased technological progress, and scale technological progress.

First, assuming that $\rho_k^t(x^{t+1}, y^{t+1}, b^{t+1})$ and $\rho_k^{t+1}(x^{t+1}, y^{t+1}, b^{t+1})$ are the efficiency of the $k_{th}$ DMU in period t to t+1, China's green Malmquist productivity index for manufacturing is defined as follows:

$$MI_k^{t,t+1} = \left[\frac{\rho_k^t(x^{t+1}, y^{t+1}, b^{t+1})}{\rho_k^t(x^t, y^t, b^t)} \times \frac{\rho_k^{t+1}(x^{t+1}, y^{t+1}, b^{t+1})}{\rho_k^{t+1}(x^t, y^t, b^t)}\right]^{\frac{1}{2}} \tag{3}$$

When $MI_k^{t,t+1} > 1$ indicates that the green TFP is growing from period t to period $t + 1$, when $MI_k^{t,t+1} < 1$ indicates that the green TFP is reduced from period t to period $t + 1$. According to the Malmquist Exponential Decomposition Method by Fare, [8] the TFP growth rate is decomposed into technological change, efficiency change as follows:

$$MI_k^{t,t+1} = TC_k^{t,t+1} \times EC_k^{t,t+1} = \left[\frac{\rho_k^t(x^t, y^t, b^t)}{\rho_k^{t+1}(x^t, y^t, b^t)} \times \frac{\rho_k^t(x^{t+1}, y^{t+1}, b^{t+1})}{\rho_k^{t+1}(x^{t+1}, y^{t+1}, b^{t+1})}\right]^{\frac{1}{2}} \times \frac{\rho_k^{t+1}(x^{t+1}, y^{t+1}, b^{t+1})}{\rho_k^t(x^t, y^t, b^t)} \tag{4}$$

$TC_k^{t,t+1}$ denotes the shift of the $k_{th}$ DMU in the period $t$ to $t + 1$ of the technological change, i.e., the technological frontier. $EC_k^{t,t+1}$ indicates a change in relative efficiency.

After breaking down MI Fare [27], decomposing TC into an index of technological progress at scale (MATC) and an index of technological bias (BTC), the technology bias index can be decomposed into input-biased technological progress and output-biased technological progress indices as follows:

$$TC_k^{t,t+1} = \left[\frac{\rho_k^t(x^t, y^t, b^t)}{\rho_k^{t+1}(x^t, y^t, b^t)} \times \frac{\rho_k^t(x^{t+1}, y^{t+1}, b^{t+1})}{\rho_k^{t+1}(x^{t+1}, y^{t+1}, b^{t+1})}\right]^{\frac{1}{2}}$$
$$= \frac{\rho_k^t(x^{t+1}, y^{t+1}, b^{t+1})}{\rho_k^{t+1}(x^{t+1}, y^{t+1}, b^{t+1})} \times \left[\frac{\rho_k^t(x^t, y^t, b^t)}{\rho_k^{t+1}(x^t, y^t, b^t)} \times \frac{\rho_k^{t+1}(x^{t+1}, y^{t+1}, b^{t+1})}{\rho_k^t(x^{t+1}, y^{t+1}, b^{t+1})}\right]^{\frac{1}{2}} \tag{5}$$
$$= MATC_k^{t,t+1} \times BTC_k^{t,t+1}$$

$$BTC_k^{t,t+1} = \left[ \frac{\rho_k^t(x^t,y^t,b^t)}{\rho_k^{t+1}(x^t,y^t,b^t)} \times \frac{\rho_k^{t+1}(x^{t+1},y^{t+1},b^{t+1})}{\rho_k^t(x^{t+1},y^{t+1},b^{t+1})} \right]^{\frac{1}{2}}$$

$$= \left[ \frac{\rho_k^{t+1}(x^t,y^t,b^t)}{\rho_k^t(x^t,y^t,b^t)} \times \frac{\rho_k^t(x^{t+1},y^t,b^t)}{\rho_k^{t+1}(x^{t+1},y^t,b^t)} \right]^{\frac{1}{2}} \times \left[ \frac{\rho_k^t(x^{t+1},y^{t+1},b^{t+1})}{\rho_k^{t+1}(x^{t+1},y^{t+1},b^{t+1})} \times \frac{\rho_k^{t+1}(x^{t+1},y^t,b^t)}{\rho_k^t(x^{t+1},y^t,b^t)} \right]^{\frac{1}{2}} \quad (6)$$

$$= IBTC_k^{t,t+1} \times OBTC_k^{t,t+1}$$

i.e.,

$$TC_k^{t,t+1} = MATC_k^{t,t+1} \times IBTC_k^{t,t+1} \times OBTC_k^{t,t+1} \quad (7)$$

MATC represents the scale effect of technological progress and is a neutral transfer of the technological frontier, while BTC means the bias of technological progress and is a "non-neutral" transfer of the technological frontier. IBTC and OBTC reflect the impact of input and output changes on technological progress. If IBTC (OBTC) >1(<1), indicates progress (regression) in input-biased technology. When IBTC and OBTC = 1, it means that the technology change is Hicks-neutral.

It needs to be pointed out that the input-biased technological change indicates the technological change range of different inputs when the output remains unchanged. Drawing on the ideas of Weber and Domazlicky [54] and Li et al. [32] on the discriminative approach to the relationship between the direction of technological change and the elements. When $x_1^{t+1}/x_2^{t+1} > x_1^t/x_2^t$ IBTC>1 indicates an $x_1$-saving/$x_2$-using biased technological change and IBTC<1 indicates an $x_1$-using/$x_2$-saving biased technological change. When IBTC = 1, the input-biased technological change is Hicks-neutral. When $x_1^{t+1}/x_2^{t+1} < x_1^t/x_2^t$ IBTC>1 indicates an $x_1$-using/$x_2$-saving biased technological change and IBTC<1 indicates an $x_1$-saving/$x_2$-using biased technological change. The $x_1$-saving/$x_2$-using biased technological change implies that the technological change is biased in favor of using more $x_2$ relative to $x_1$, while the $x_1$-using/$x_2$-saving biased technological change implies that the technological change is biased in favor of using more $x_1$ relative to $x_2$.

The principle of OBTC is similar to that of IBTC as described above. When $y_1^{t+1}/y_2^{t+1} > y_1^t/y_2^t$ OBTC>1 indicates a $y_2$-producing biased technological change and OBTC<1 indicates a $y_1$-producing biased technological change. When OBTC = 1, the output-biased technological change is Hicks-neutral. When $y_1^{t+1}/y_2^{t+1} < y_1^t/y_2^t$ OBTC>1 indicates a $y_1$-producing biased technological change and OBTC<1 indicates a $y_2$-producing biased technological change. The $y_1$-producing biased technological change means that output-biased technology tends to produce more $y_1$ relative to $y_2$. While the $y_2$-producing biased technological change tends to produce more $y_2$ relative to $y_1$.

Specific descriptions of technical bias relationships are listed in Table 1. $x_l$ represents labor factor inputs, $x_k$ represents industry capital investment, $x_e$ represents energy factor inputs; $y_g$ represents desired output; $y_b$ represents the undesired output.

In general, there are two main conclusions about the impact of environmental regulations on technological progress. One view is that environmental regulations increase production costs and affect R&D investment, which in turn affects firm performance and productivity. Another view is that when the intensity of environmental regulation is low, firms tend to pay certain environmental taxes and emissions costs rather than research and development on green technologies and that emissions costs have a crowding-out effect on research and development technologies. When environmental regulations reach a certain level of intensity, the increase in environmental taxes and emissions costs leads to rising production costs, which in turn leads to technological innovation in the direction of cleaner technologies, just as the environmental Kuznets curve depicts the relationship between the level of economic development and environmental quality. Therefore, when investigating the mechanism of the influence of environmental regulations on technological

progress, this paper considers applying a threshold model to study it as the following empirical model:

$$
\begin{aligned}
LncTC_{it} = {} & \alpha_0 + \alpha_1 ER_{it} \times I(ER_{it} \leq \gamma_1) + \alpha_2 ER_{it} \times I(\gamma_1 < ER_{it} \leq \gamma_2) + \\
& \alpha_3 ER_{it} \times I(ER_{it} > \gamma_2) + \beta_1 LnPROP_{it} + \beta_2 LnR\&D_{it} + \beta_3 LnEDV_{it} \\
& + \beta_4 LnASE_{it} + \beta_5 LnSEC_{it} + \varepsilon_{it}
\end{aligned}
\tag{8}
$$

TC contains TC and breaks down items like BTC IBTC OBTC technical bias indicators. For the sake of space, only the TC model is listed here, *i* representing Industries, *t* representing years, $\varepsilon_{it}$ is random perturbation terms. $ER_{it}$ is for the environmental regulation intensity variable, $LnPROP_{it}$, $LnR\&D_{it}$, $LnEDV_{it}$, $LnASE_{it}$, $LnSEC_{it}$ are the control variables for R&D intensity of ownership structure, industry exports, average industry size, and energy consumption structure by industry, respectively.

**Table 1.** Biased technical change direction in input and output mix.

| Input mix | IBTC>1 | IBTC=1 | IBTC<1 |
|---|---|---|---|
| $\dfrac{x_k^{t+1}}{x_l^{t+1}} > \dfrac{x_k^t}{x_l^t}$ | $x_k$-saving, $x_l$-using | Neutral | $x_l$-saving, $x_k$-using |
| $\dfrac{x_k^{t+1}}{x_l^{t+1}} < \dfrac{x_k^t}{x_l^t}$ | $x_l$-saving, $x_k$-using | Neutral | $x_k$-saving, $x_l$-using |
| $\dfrac{x_k^{t+1}}{x_e^{t+1}} > \dfrac{x_k^t}{x_e^t}$ | $x_k$-saving, $x_e$-using | Neutral | $x_e$-saving, $x_k$-using |
| $\dfrac{x_k^{t+1}}{x_e^{t+1}} < \dfrac{x_k^t}{x_e^t}$ | $x_e$-saving, $x_k$-using | Neutral | $x_k$-saving, $x_e$-using |
| $\dfrac{x_l^{t+1}}{x_e^{t+1}} > \dfrac{x_l^t}{x_e^t}$ | $x_l$-saving, $x_e$-using | Neutral | $x_e$-saving, $x_l$-using |
| $\dfrac{x_l^{t+1}}{x_e^{t+1}} < \dfrac{x_l^t}{x_e^t}$ | $x_e$-saving, $x_l$-using | Neutral | $x_l$-saving, $x_e$-using |
| **Output mix** | $OBTC > 1$ | $OBTC = 1$ | $OBTC < 1$ |
| $\dfrac{y_b^{t+1}}{y_g^{t+1}} > \dfrac{y_b^t}{y_g^t}$ | Promote desirable output | Neutral | Increase undesirable output |
| $\dfrac{y_b^{t+1}}{y_g^{t+1}} < \dfrac{y_b^t}{y_g^t}$ | Increase undesirable output | Neutral | Promote desirable output |

### 3.2. Data and Sources

In this paper, we use the panel data of Chinese manufacturing industries to measure the green TFP and examine technological progress bias. Based on the completeness and reliability of the manufacturing data and the availability of influencing factor data required for the following analysis, we set the research time span as 2003–2015. Based on input-output data from 27 manufacturing industries, the SBM model with undesired output and the MI index are used to measure the growth rate of green TFP in China's manufacturing industry, which is decomposed into input-biased technological progress, output-biased technological progress, technological scale change, and technological efficiency change. Input/output data and how they are processed are as follows:

1. Labor input. The labor force data is measured by using the average annual number of employees of enterprises above designated size in 27 subindustries of China's manufacturing industry, which is taken from the "China Industrial Economic Statistics Yearbook."

2. Capital investment. The average annual balance of net fixed assets of enterprises above designated size in 27 subindustries in China's manufacturing industry is used as an approximate estimate of the capital stock, and the fixed asset investment price index of each industry is converted into the constant price in 2000.

3. Energy input. The total energy consumption of enterprises above designated size in 27 subindustries in China's manufacturing industry is measured by the data from the "China Energy Statistical Yearbook," which is converted into 10,000 tons of standard coal according to the conversion factor from the attached list in the "China Energy Statistical Yearbook."

4.  Expected Output. The expected output is expressed using the main business income of above-scale enterprises in 27 subindustries of China's manufacturing industry, with price index deflations using 2000 as the base period.
5.  Undesired Output. Using the calculation method of carbon emissions in the Guidelines for National Greenhouse Gas Inventories compiled by the Intergovernmental Panel on Climate Change (IPCC, 2016), the carbon emissions of enterprises above designated size by industry are obtained by summing the estimates using coal, coke, crude oil, gasoline, kerosene, diesel, fuel oil, and natural gas as benchmarks.

The formula is:

$$CO_2 = \sum_{i=1}^{8} CO_{2,it} = \sum_{i=1}^{8} E_{it} \times NCV_i \times CEF_i$$

Among them, $CO_2$ represents the amount of carbon dioxide emissions to be estimated, $i$ represents different types of energy, $E_{it}$ represents the combustion consumption of various energy sources, and $NCV_i$ represents the average low calorific value of various energy sources. The value comes from the "China Energy Statistical Yearbook". $CEF_i$ represents the carbon dioxide emission factor of various energy sources, and the value comes from IPCC (2016).

In order to identify the factors influencing the technological progress bias, this paper also includes variables such as Environmental regulation, Proportion of state-owned Assets, R&D investment, Export delivery value, Average size of enterprises, Structure of energy consumption in the analysis.

Environmental regulation (ER): Environmental regulations can represent a country's need for greening. The intensity of environmental regulations is expressed as ER using the current year operating costs of wastewater treatment facilities and the operating expenses of exhaust gas treatment facilities in each industry.

The proportion of state-owned Assets (PROP): Given that a large proportion of Chinese manufacturing enterprises are either state-owned or collectively owned, the choice of environmental protection varies significantly among different types of enterprises, so the impact of the ownership structure on the environment cannot be ignored, and the proportion of state-owned assets is used to characterize this variable.

R&D investment (R&D): This variable represents the actual innovation intensity of the industry and is measured by the R&D expenditure.

Export delivery value (EDV): The development of international trade can expand the market scale of China's domestic enterprises. Chinese enterprises can improve the level of green technology through export learning and technology import, and this variable is represented by the export value of the industry.

The average size of enterprises (ASE): Firms with greater monopoly power in the market are able to invest continuously in innovation and thus maintain a stronger capacity for technological innovation. The larger the size of an enterprise, the higher its profitability and market position will be, so the average size of the industry is used to characterize this variable.

Structure of energy consumption (SEC): Coal consumption as a percentage of total consumption is a measure to the structure of energy consumption in an industry, with higher levels of this variable indicating that the industry is less green. Coal consumption, as a percentage of total energy consumption, is used to characterize the structure of the energy consumption variable.

The data for the above industry economic variables are from China Industrial Statistical Yearbook, the data for environmental regulations are from China Environmental Statistical Yearbook, and the data for energy are from China Energy Statistical Yearbook. Descriptive statistics for each of the above indicators are shown in Table 2.

## 4. Empirical Results and Discussion

### 4.1. The Green TFP Growth and Technological Bias in China's Manufacturing Sector

In order to better understand the sources of green TFP growth and technology change in the Chinese manufacturing sector from 2003 to 2015, Table 3 presents the average of green TFP and its decomposition over the observation period for 27 industries in the Chinese manufacturing sector. According to the literature review, the current research on the details of China's manufacturing technology progress is not sufficient [29,32,50,51]. The interpretation of the results can effectively enable us to better understand the current situation of China's manufacturing industry and pave the way for the following environmental regulations.

**Table 2.** The descriptive statistics of China's manufacturing sector's input and output indicators during 2003–2015.

|  | Variable | Description of Variable | Unit | Min | Max | Average | SD |
|---|---|---|---|---|---|---|---|
| Inputs and Outputs | L | Labor | Ten thousand-person | 18.610 | 909.260 | 262.219 | 183.058 |
|  | K | Capital stock | 100 million yuan | 146.400 | 57,316.290 | 4218.415 | 6620.155 |
|  | E | Energy consumption | Ten thousand tons | 109.380 | 69,342.000 | 7493.792 | 13,175.540 |
|  | MBI | Main business income | 100 million yuan | 510.522 | 77,389.360 | 15,784.550 | 15,952.750 |
|  | $CO_2$ | Carbon dioxide emission | Ten thousand tons | 66.537 | 376,910.800 | 22,802.300 | 61,013.560 |
| Influencing Factors | ER | Environmental regulation | | 0.027 | 0.052 | 0.036 | 0.004 |
|  | PROP | Proportion of state-owned Assets | | 0.008 | 0.993 | 0.282 | 0.253 |
|  | R & D | R&D investment | | 0.001 | 0.027 | 0.009 | 0.006 |
|  | EDV | Export delivery value | | 21.230 | 46,165.140 | 2824.249 | 6052.567 |
|  | ASE | Average size of enterprises | | 0.221 | 70.232 | 3.458 | 8.093 |
|  | SEC | Structure of energy consumption | | 0.153 | 0.980 | 0.653 | 0.235 |

From the mean results over the sample period, although the overall green TFP growth rate in China's manufacturing industry is always greater than 1, the growth rate is declining, with an average annual growth rate of −0.622%. From the decomposition term, technical efficiency EC received an average annual growth rate of −0.105%, and technical progress TC received an average yearly growth rate of −0.497%. Compared with efficiency change, technical change is the leading cause of green TFP regression, accounting for 79.9% of the contribution to green TFP regression.

Figure 1 shows the change of green TFP and its decomposition term in the Chinese manufacturing industry from 2003 to 2015. From Figure 1, combined with Table 3, we can see that the green TFP growth index (MI) of China's manufacturing industry is always greater than 1, indicating that its overall green TFP is growing. Still, the growth rate is on the whole declining, and the trend is roughly the same as that of TC. It shows that the regression in the growth rate of green TFP is mainly caused by technological regression. From the diagram of the TC decomposition term, we can see that the trend of MATC indicator, i.e., technology scale change, is closer to the TFP trend than the bias indicator, indicating that the green TFP change in the Chinese manufacturing industry is more influenced by the size of technology scale change. This phenomenon shows that Chinese manufacturing enterprises are small in size and large in number and have not achieved economies of scale, which is consistent with the reality of the Chinese manufacturing industry.

To better understand the bias of green TFP growth and technological progress in China's manufacturing sector during 2003–2015, the green TFP and its decomposition, TC and its decomposition, and BTC and its decomposition are multiplied cumulatively, as shown in Figure 2. From Figure 2, the trends in green TFP and its decomposition terms EC and TC again demonstrate that green TFP growth during the observation period was dominated by technological progress rather than efficiency gains. From the cumulative trend of the TC and its decomposition term, the impact of scaled technological change on technological progress is important. This result is consistent with previous findings in Figure 1 and indicates that the development of China's manufacturing sector relies heavily on the expansion of firm size to increase output and efficiency. From the cumulative trend of BTC and its decomposition term, the technology bias (BTC) index is permanently greater than 1 during the observation period and shows an increasing trend, indicating that the bias of technological progress from 2003 to 2015 promoted green TFP in Chinese manufacturing. Combined with Figures 1 and 2, the technology bias grew sluggishly during 2007–2009,

and the TFP growth rate experienced a downward trend during this period, which is related to the global economic crisis that broke out during this period. Since 2010, when the Chinese government issued a series of measures such as the Targeted Responsibility System for Energy Conservation and Emission Reduction for local governments at all levels and the Action Plan for Prevention and Control of Air Pollution in 2013, the growth rate of green TFP in China's manufacturing industry slowed down during 2011–2015. This is due to the tough environmental regulations that the industry faces as it grows. Environmental regulations force the development of green technologies, while outdated production capacities and technologies face obsolescence. Different types of enterprises show several choices when faced with the green technology transition. SOEs or well-capitalized enterprises may go through the "pains" of technology transition and invest a large amount of capital in research and development of green production technologies to achieve the goal of energy-saving and emission reduction. However, in this period of technological change, some companies that have not developed and applied green technologies are facing a crisis of obsolescence, which is also the growth rate curve of MATC. The reason for the fluctuation during this period. The BTC index and its decomposition went from flat to rising during the period 2011–2015, indicating that China's manufacturing sector's technological progress was biased toward green growth during this period.

**Table 3.** The geometric means of green TFP of China's manufacturing sector and its decomposition from 2003 to 2015.

| Sector | MI | | EC | | TC | | BTC | | IBTC | | OBTC | | MTC | |
|---|---|---|---|---|---|---|---|---|---|---|---|---|---|---|
| | Value | SD | Value | SD | Value | SD | Value | SD | Value | SD | Value | SD | Value | SD |
| Processing of Food from Agricultural Products | 1.062 | 0.086 | 1.041 | 0.098 | 1.022 | 0.053 | 1.001 | 0.003 | 1.011 | 0.030 | 0.990 | 0.028 | 1.021 | 0.054 |
| Manufacture of Foods | 1.073 | 0.061 | 1.020 | 0.042 | 1.052 | 0.038 | 1.001 | 0.006 | 1.000 | 0.007 | 1.002 | 0.004 | 1.051 | 0.039 |
| Manufacture of Liquor, Beverages and Refined Tea | 1.099 | 0.058 | 1.025 | 0.040 | 1.072 | 0.043 | 1.001 | 0.007 | 1.001 | 0.007 | 1.001 | 0.005 | 1.070 | 0.039 |
| Manufacture of Tobacco | 1.019 | 0.033 | 1.032 | 0.056 | 0.990 | 0.044 | 0.992 | 0.011 | 1.005 | 0.016 | 0.987 | 0.017 | 0.998 | 0.046 |
| Manufacture of Textile | 1.070 | 0.037 | 1.028 | 0.054 | 1.043 | 0.056 | 1.003 | 0.005 | 1.000 | 0.010 | 1.003 | 0.008 | 1.041 | 0.057 |
| Manufacture of Textile, Wearing Apparel and Accessories | 1.047 | 0.070 | 0.990 | 0.053 | 1.061 | 0.100 | 1.013 | 0.018 | 0.999 | 0.008 | 1.013 | 0.018 | 1.047 | 0.090 |
| Manufacture of Leather, Fur, Feather and Related Products and Footwear | 1.009 | 0.030 | 0.994 | 0.136 | 1.036 | 0.177 | 1.003 | 0.040 | 1.026 | 0.064 | 0.980 | 0.055 | 1.039 | 0.210 |
| Processing of Timber, Manufacture of Wood, Bamboo, Rattan, Palm and Straw Products | 1.045 | 0.070 | 1.012 | 0.048 | 1.034 | 0.067 | 1.011 | 0.015 | 1.003 | 0.013 | 1.008 | 0.012 | 1.024 | 0.077 |
| Manufacture of Furniture | 0.946 | 0.074 | 0.955 | 0.117 | 0.999 | 0.078 | 1.010 | 0.043 | 0.993 | 0.050 | 1.017 | 0.030 | 0.994 | 0.122 |
| Manufacture of Paper and Paper Products | 1.109 | 0.058 | 1.016 | 0.065 | 1.094 | 0.066 | 1.001 | 0.005 | 0.998 | 0.008 | 1.003 | 0.008 | 1.093 | 0.066 |
| Printing and Reproduction of Recording Media | 1.001 | 0.260 | 1.013 | 0.248 | 0.991 | 0.088 | 1.022 | 0.054 | 0.989 | 0.048 | 1.035 | 0.049 | 0.970 | 0.079 |
| Manufacture of Articles for Culture, Education, Arts and Crafts, Sport and Entertainment Activities | 1.000 | 0.070 | 1.002 | 0.057 | 0.998 | 0.030 | 1.006 | 0.007 | 1.000 | 0.027 | 1.007 | 0.028 | 0.992 | 0.025 |
| Processing of Petroleum, Coking and Processing of Nuclear Fuel | 1.017 | 0.013 | 1.001 | 0.013 | 1.016 | 0.006 | 1.001 | 0.003 | 1.001 | 0.001 | 1.000 | 0.003 | 1.015 | 0.007 |
| Manufacture of Raw Chemical Materials and Chemical Products | 1.176 | 0.085 | 1.080 | 0.081 | 1.092 | 0.089 | 0.997 | 0.051 | 0.998 | 0.010 | 0.999 | 0.052 | 1.097 | 0.092 |
| Manufacture of Medicines | 1.084 | 0.078 | 1.022 | 0.065 | 1.062 | 0.048 | 1.002 | 0.006 | 1.002 | 0.006 | 1.000 | 0.005 | 1.060 | 0.046 |
| Manufacture of Chemical Fibers | 1.093 | 0.124 | 0.993 | 0.121 | 1.104 | 0.064 | 1.018 | 0.014 | 1.007 | 0.013 | 1.011 | 0.014 | 1.084 | 0.066 |
| Manufacture of Rubber and Plastics Products | 1.071 | 0.035 | 1.020 | 0.038 | 1.051 | 0.050 | 1.003 | 0.007 | 1.000 | 0.009 | 1.003 | 0.005 | 1.048 | 0.050 |
| Manufacture of Non-metallic Mineral Products | 1.13 | 0.050 | 1.054 | 0.069 | 1.079 | 0.072 | 1.001 | 0.010 | 0.997 | 0.007 | 1.004 | 0.012 | 1.078 | 0.075 |
| Smelting and Pressing of Ferrous Metals | 1.098 | 0.182 | 1.050 | 0.181 | 1.050 | 0.078 | 0.987 | 0.030 | 0.992 | 0.031 | 0.995 | 0.035 | 1.066 | 0.103 |
| Smelting and Pressing of Non-ferrous Metals | 1.121 | 0.184 | 1.088 | 0.172 | 1.030 | 0.044 | 0.998 | 0.008 | 1.010 | 0.030 | 0.989 | 0.027 | 0.998 | 0.008 |
| Manufacture of Metal Products | 1.064 | 0.050 | 1.008 | 0.051 | 1.057 | 0.052 | 1.005 | 0.007 | 1.001 | 0.008 | 1.004 | 0.006 | 1.052 | 0.053 |
| Manufacture of General Purpose Machinery | 1.088 | 0.047 | 1.033 | 0.052 | 1.055 | 0.057 | 1.001 | 0.008 | 0.996 | 0.008 | 1.005 | 0.008 | 1.053 | 0.055 |
| Manufacture of Special Purpose Machinery | 1.084 | 0.028 | 1.028 | 0.057 | 1.057 | 0.051 | 0.998 | 0.007 | 0.997 | 0.005 | 1.001 | 0.006 | 1.058 | 0.048 |
| Manufacture of Transport Equipmem | 1.075 | 0.096 | 1.051 | 0.085 | 1.027 | 0.109 | 0.999 | 0.037 | 1.007 | 0.034 | 0.992 | 0.038 | 1.030 | 0.114 |
| Manufacture of Electrical Machinery and Apparatus | 1.081 | 0.068 | 1.018 | 0.078 | 1.065 | 0.077 | 0.993 | 0.031 | 0.997 | 0.026 | 0.996 | 0.032 | 1.074 | 0.092 |
| Manufacture of Computers, Communication and Other Electronic Equipment | 1.022 | 0.018 | 1.006 | 0.026 | 1.016 | 0.020 | 1.002 | 0.004 | 1.001 | 0.002 | 1.001 | 0.004 | 1.014 | 0.019 |
| Manufacture of Measuring Instruments and Machinery | 1.034 | 0.074 | 1.003 | 0.040 | 1.031 | 0.062 | 1.007 | 0.039 | 1.011 | 0.034 | 0.996 | 0.038 | 1.027 | 0.102 |
| Average | 1.064 | 0.076 | 1.022 | 0.079 | 1.044 | 0.064 | 1.003 | 0.018 | 1.002 | 0.019 | 1.002 | 0.020 | 1.041 | 0.068 |

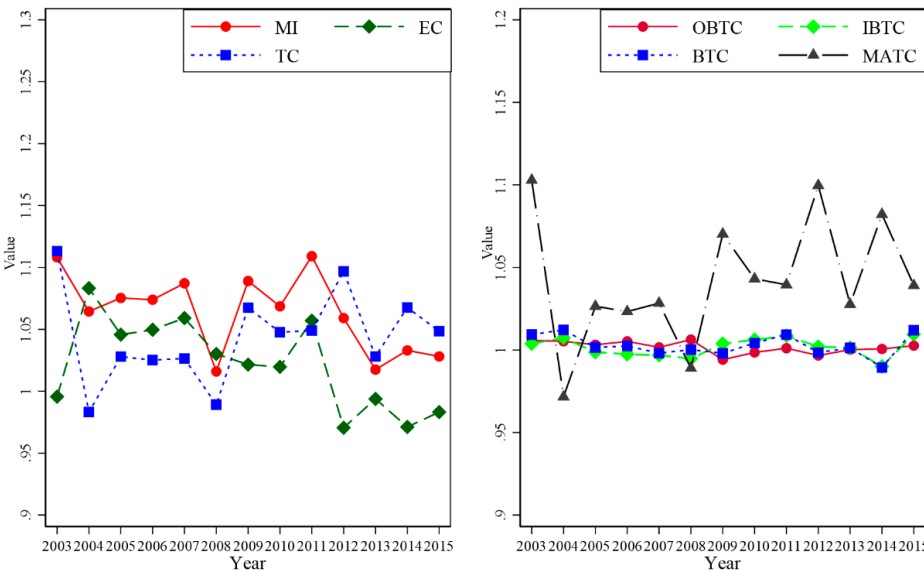

**Figure 1.** Green TFP in China's manufacturing sector and it's decomposition: China's green Malmquist productivity index (MI), efficiency change (EC), technological change (TC), output-biased technological change (OBTC), input-biased technological change (IBTC), bias technological change (BTC), magnitude of technological change (MATC).

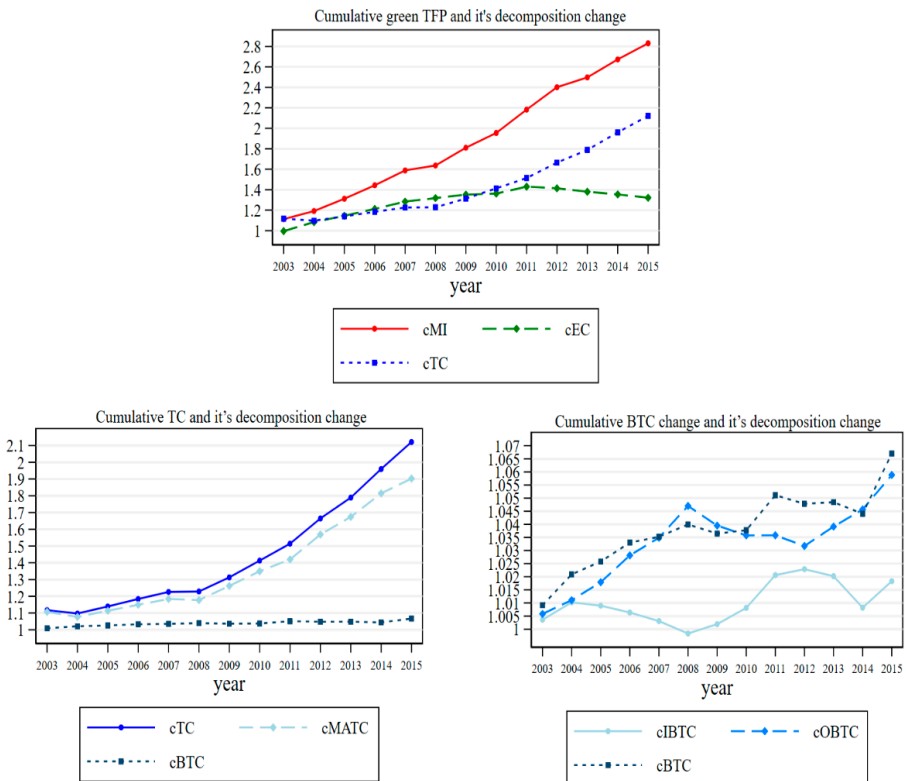

**Figure 2.** Cumulative green TFP change and its decomposition in China's manufacturing sector.

From the previous analysis, we can see that technological progress is an important component of TFP and dominates the growth trend of green TFP in China's manufacturing industry. A technological progress indicator is composed of scale technological progress and biased technological progress. Biased technological progress represents the directionality of the production technology when BTC = 1, i.e., when both IBTC and OBTC are equal to 1, indicating a Hicks-neutral technological change at this time. The results in Table 2

show that the mean value of the BTC index is 1.003, which is very close to 1, indicating that the overall technological progress in China's manufacturing sector during the sample period was neutral technological change. However, the maximum, mean, and minimum values of BTC, IBTC, and OBTC for each year shown in Figure 3 show that the bias of technological progress in China's manufacturing industry deviates greatly from industry to industry.

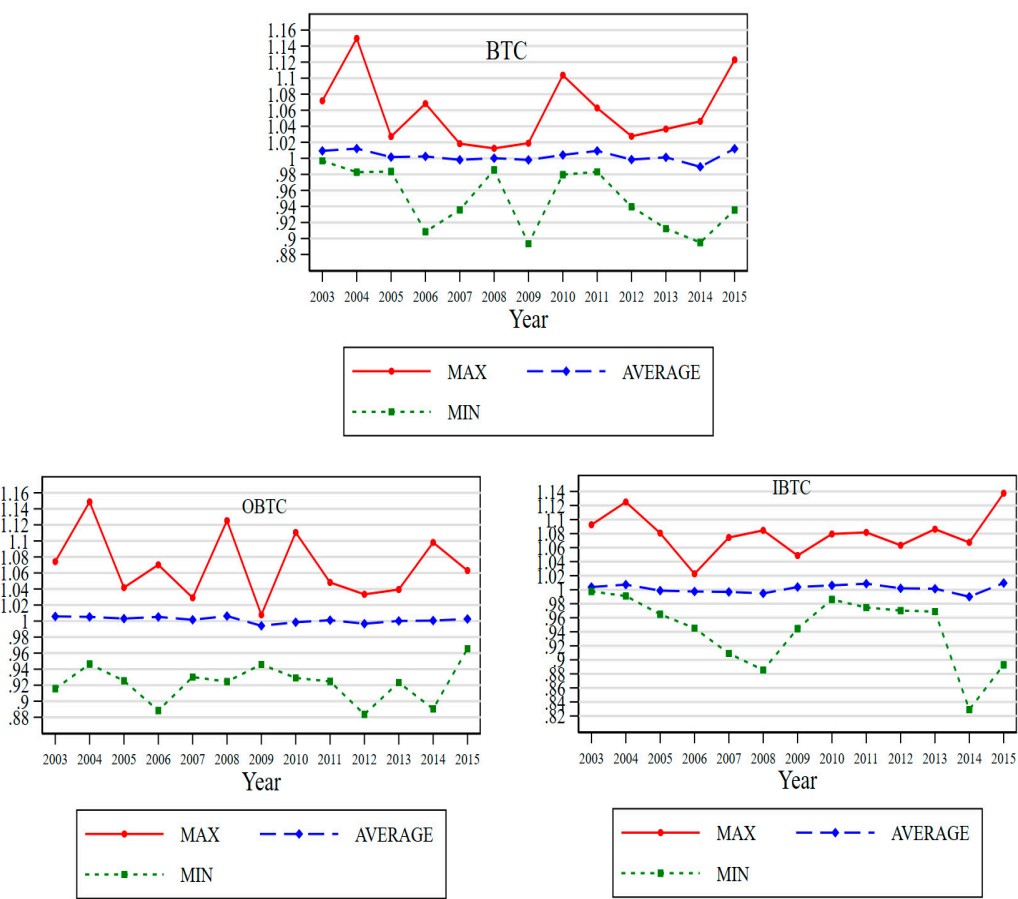

**Figure 3.** The maximum and minimum of BTC values and its decomposition.

As shown in Figure 3, the trends and sizes of the maximum and minimum values are very different. This shows that with time changes, the specific details of technological progress fluctuate greatly. Both input bias and output bias of technological progress bias are significantly higher than in others in some industries. A higher technology bias can lead to higher input and output efficiency, driving TFP growth in the industry effectively. However, here is not to emphasize the development of a specific industry, but to explore the relationship between such fluctuations and environmental regulations based on subsequent analysis. In terms of mean values, the change in BTC is not significant. Still, the large variability in technology bias in BTC between industries, especially the fit between input-output ratios and technology progress bias, deserves further discussion.

### 4.2. The Direction of the Green-Biased Technological Change

It is worth noting that IBTC and OBTC reflect only TFP growth rates and not the technological progress associated with a particular input or output bias. As shown in Section 3, changes in IBTC and OBTC can either promote or reduce TFP growth. Changes in the mix of inputs and outputs can lead to a bias in technological change. Based on what we have studied in this paper, we analyze the bias between the input-output mix and

technological progress for each year of the Chinese manufacturing subindustry, and the resulting observations are shown in Table 4.

From the results of the analysis, it can be seen that all industries in China's manufacturing sector have a bias for technological progress and a bias for input-output. With the mind of Li, [32] classifying the observation periods according to China's 10th Five-Year Plan (2001–2005), 11th Five-Year Plan (2006–2010), and 12th Five-Year Plan (2011–2015), it can be seen that the factor bias of China's manufacturing technology progress has the following characteristics.

**Table 4.** Distributions of annual input-biased and output-biased technological changes.

| Year | IBTC | | OBTC | | K vs. L | | K vs. E | | L vs. E | | Desirable Output vs. $CO_2$ | |
|---|---|---|---|---|---|---|---|---|---|---|---|---|
| | >1 | <1 | >1 | <1 | K-Using | L-Using | K-Using | E-Using | L-Using | E-Using | Desirable Output-Producing | $CO_2$-Producing |
| 2003 | 13 | 14 | 21 | 6 | 20 | 7 | 10 | 17 | 13 | 14 | 6 | 21 |
| 2004 | 10 | 17 | 21 | 6 | 14 | 13 | 7 | 20 | 9 | 18 | 15 | 12 |
| 2005 | 7 | 20 | 21 | 6 | 18 | 9 | 11 | 16 | 6 | 21 | 7 | 20 |
| 2006 | 9 | 18 | 22 | 5 | 19 | 8 | 12 | 15 | 7 | 20 | 5 | 22 |
| 2007 | 7 | 20 | 18 | 9 | 18 | 9 | 22 | 5 | 16 | 11 | 9 | 18 |
| 2008 | 11 | 16 | 18 | 9 | 16 | 11 | 19 | 8 | 17 | 10 | 9 | 18 |
| 2009 | 17 | 10 | 10 | 17 | 10 | 17 | 10 | 17 | 10 | 17 | 18 | 9 |
| 2010 | 12 | 15 | 12 | 15 | 16 | 11 | 11 | 16 | 11 | 16 | 14 | 13 |
| 2011 | 18 | 9 | 15 | 12 | 8 | 19 | 12 | 15 | 18 | 9 | 12 | 15 |
| 2012 | 12 | 15 | 12 | 15 | 13 | 14 | 15 | 12 | 12 | 15 | 15 | 12 |
| 2013 | 9 | 18 | 20 | 7 | 17 | 10 | 18 | 9 | 22 | 5 | 20 | 7 |
| 2014 | 14 | 13 | 19 | 8 | 14 | 13 | 12 | 15 | 17 | 10 | 10 | 17 |
| 2015 | 19 | 8 | 13 | 14 | 8 | 19 | 6 | 21 | 10 | 17 | 17 | 10 |
| 10thFYP | >1 | | >1 | | L | | K | | L | | $CO_2$ | |
| 11thFYP | <1 | | >1 | | K | | K | | L | | $CO_2$ | |
| 12thFYP | >1 | | >1 | | L | | E | | L | | $CO_2$ | |

Chronologically, the Chinese manufacturing industry is characterized by an L-K-L factor bias between capital and labor, which is related to the fact that the Chinese manufacturing industry starts with labor-intensive production methods. The factor bias between labor and energy is also confirmed by the fact that capital is scarcer than labor in China at a relatively cheap level.

During the observation period, there is an L-L-L bias between labor and energy in China's manufacturing industry, which is also due to the relatively abundant supply of labor compared to energy, and the fact that technological progress is biased toward labor factors, which can progress the TFP growth and is more suitable for China's national conditions.

There is a K-K-E bias between capital and energy, indicating a gradual approach from a capital bias to an energy bias in the development of China's manufacturing sector. China is a relatively energy-poor country, relying on imports for most of its energy and relying more on capital for rapid TFP growth at the beginning of the observation period. The results of this analysis reinforce the importance of green energy for the development of China's manufacturing sector.

As for the output bias in Table 4, although the number of industries with output bias toward the desired output is increasing, the number of sectors with $CO_2$ bias is decreasing from 2003 to 2015 with a trend of environmental protection and emission reduction. However, China's manufacturing industry as a whole has been biased towards the generation of undesirable output $CO_2$ during this period. This may be due to the fact that China's manufacturing industry was still in a phase of rapid development during the study period, with companies focusing more on expanding scale and increasing production capacity than on energy conservation and environmental protection.

It can be seen that although China has carried out a series of environmental regulatory policies and support policies for green technology development, and the emission situation of related industries has improved, the actual situation of energy conservation and emission

reduction in China is still not satisfactory. The results of this analysis are more consistent with the actual situation of China's manufacturing development.

### 4.3. The Influencing Factors of Technological Bias and It's Threshold Model

After completing the analysis of China's manufacturing technology progress TC and its decomposition items IBTC and OBTC, we use the above panel data to examine the factors affecting China's manufacturing technology progress and its bias. It should be pointed out that, considering the need for continuity and stability of the technological bias here, we adopt the cumulative technological advancement cTC, the accumulated input biased technological advancement cIBTC, and the accumulated output biased technological advancement cOBTC, and logarithmic all indicators.

According to the threshold regression principle, environmental regulations are first made as threshold variables to fit the relationship between environmental regulations and technological progress. To determine the specific form of the measurement model, it is necessary to determine the number of thresholds for environmental regulation and the corresponding thresholds. The model (8) is estimated under the assumption that there is no threshold value, one threshold value, two threshold values, and three threshold values, and the corresponding F statistics can be obtained. Using the Bootstrap proposed by Hansen to repeatedly sample 300 times, the corresponding P value was obtained by simulation, and the related inspection results are shown in Table 5.

**Table 5.** Threshold test effect.

| The Threshold Test | LncIBTC | | LncOBTC | | LncBTC | | LncTC | |
|---|---|---|---|---|---|---|---|---|
| | Single Threshold | Double Threshold | Single Threshold | Double Threshold | Single Threshold | Double Threshold | Single Threshold | Double Threshold |
| F-Value | 34.310 *** | 22.800 *** | 20.030 *** | 11.87 *** | 25.160 ** | 12.71 * | 20.77 *** | 6.22 |
| *p*-Value | 0.004 | 0.000 | 0.000 | 0.003 | 0.003 | 0.0867 | 0.000 | 0.53 |
| 1% | 30.213 | 15.839 | 16.046 | 9.89 | 29.47 | 32.7891 | 12.6947 | 15.013 |
| 5% | 27.964 | 13.394 | 14.531 | 8.758 | 23.463 | 15.0074 | 10.3231 | 10.881 |
| 10% | 26.553 | 11.595 | 13.156 | 7.864 | 20.977 | 11.8642 | 8.6902 | 9.594 |

Note: ***, **, and * denote statistical significance at the 1%, 5%, and 10% levels, respectively.

Table 5 shows that if the F value of the single threshold effect test of the LncIBTC indicator is 34.31 > 30.213, the corresponding p-value is 0.004, indicating that LncIBTC rejects the null hypothesis that there is no threshold effect at the 1% significance level, and accepts the existence of a single threshold. In the same way, the double threshold effect results show that the null hypothesis that there is only one threshold value is rejected at the 1% significance level. The threshold test results of LncOBTC, LnBTC, and LncTC are shown in Table 5. Because the hypothesis of the triple threshold value of all indicators failed the significance test, it is not listed in Table 5.

After the threshold model test, the "grid search method" is used to determine the threshold value. The 95% confidence interval of the threshold value is the interval formed by the critical value at the significance level of the likelihood ratio statistic LR value less than 5%. When the confidence interval is less than the 5% significance level of the LR value, the estimated threshold value is valid. Therefore, the required likelihood ratio function graph constructed here can more intuitively show the estimation of the threshold value and the construction process of the confidence interval. As shown in Figure 4, the threshold model's LR values are all below the critical line, indicating the true validity of the model's threshold value.

The test result of the threshold model shows that environmental regulations have a threshold effect on China's manufacturing industry's technological progress. There is a double threshold effect in the biased technological progress indicators. This phenomenon shows that the impact of environmental regulations on technological progress is constrained. In other words, the environmental regulation policy may have a positive impact on the development of green technology in the early stage of implementation. However, the

intensity of environmental supervision exceeds a certain level, the benefits of continuous investment in environmental technologies may be less than the costs of applying non-green technologies. This also confirms the U-shaped relationship between environmental regulation and economic development in related studies indirectly. [19,55] Moreover, the dual-threshold effect of biased technological progress also shows that environmental regulation's influence mechanism on technological progress is not a simple linear relationship. The influence relationship between the two thresholds may be uncertain. The test results in Table 5 show that the biased technological advancement BTC and its decomposition items IBTC and OBTC both have double thresholds, but TC only has a single threshold. This phenomenon may result from the environmental regulations, and other influencing factors have different influence mechanisms on input and output-based technological progress. The change of input-biased technological progress reduces a certain element input (IBTC) based on a specific output. Output-biased technological progress relates to changes in the marginal productivity of output items based on certain inputs. Therefore, the threshold effects of IBTC and OBTC offset each other, resulting in differences in the threshold effects of BTC and TC indicators.

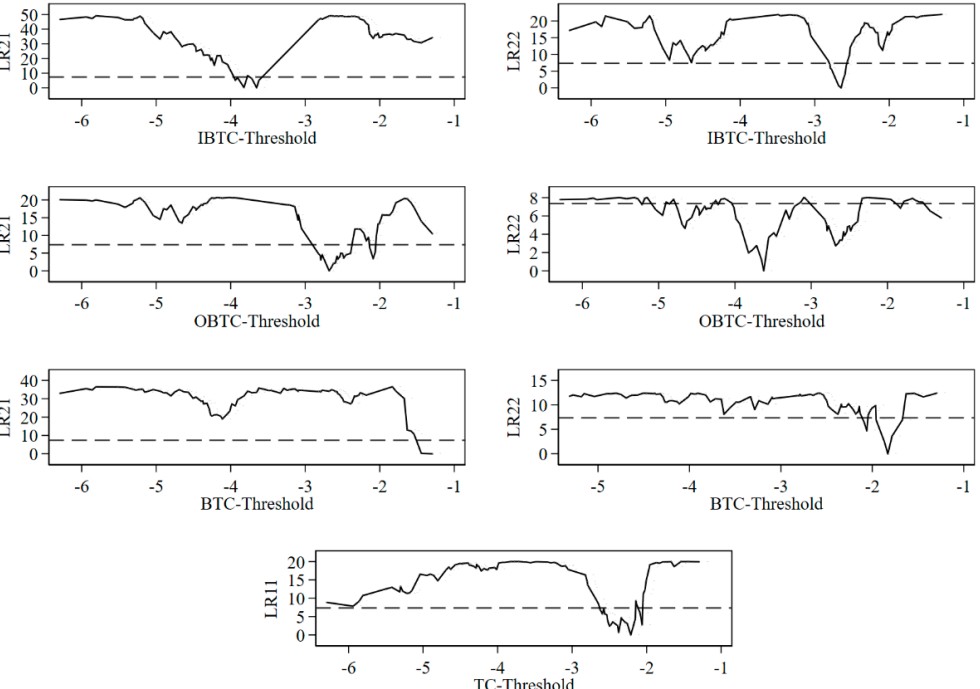

**Figure 4.** Threshold value.

As shown by the estimated results of the threshold model in Table 6, for IBTC, when LnER $\leq -3.822$, the coefficient is 0.003, and it is significant at the 1% level, indicating that under this level of environmental regulation, it can affect investment-oriented technological progress tends to produce positive effects. When environmental regulations are greater than this level, environmental regulations will have a negative effect on investment-based technological progress. As the intensity of environmental regulations increases, their negative effects become more obvious. For OBTC, it is contrary to IBTC. When LnER $\leq -3.6218$, the coefficient is $-0.019$, and it is significant at the 1% level, indicating that under this level of environmental regulation, environmental regulation is biased towards output-oriented technological progress. A negative impact occurs. When the intensity of environmental regulation rises to $-3.6218 < \text{lnER} \leq -2.6790$, the coefficient turns from negative to positive, but the significance level is not passed. However, it can still be seen that environmental regulation has an effect on output-oriented technological progress. The bias plays a promoting role. When lnER $> -2.6790$, it turns into a negative role again. Under the combined effect of the two, environmental regulations have always had a negative impact on the biased

technological progress of BTC. Still, the magnitude of the impact varies with the intensity of environmental regulations as shown in Table 6, when lnER≤−1.8297, environmental regulations have a negative effect on BTC. The impact is small. When −1.8297 < lnER ≤ −1.2931, the impact coefficient increases, and with the continuous increase of environmental regulations, the impact coefficient decreases to −0.025. As far as the control variables are concerned, the coefficients of IBTC and OBTC are mostly opposite. The nature of IBTC and OBTC also determines this. Due to space limitations, only the influence of the control variables on the overall technological progress (TC) is analyzed here. For TC, except for the property rights structure (PROP), the other influencing factors are significant at the 1% level. Among them, R&D intensity, industry experts, and energy structure have a positive impact on technological progress, and industrial scale has a negative impact. Increasing the R&D intensity can improve the level of enterprise innovation. The level of innovation and the increase in the export value of the industry can improve the technological level of the industry through export learning and technology introduction. The energy structure represents the proportion of the industry's coal consumption to the industry's energy consumption. The higher this indicator, the higher the industry's demand for fossil energy. The energy structure can promote the industry's technological progress, as described in Section 4.2 China's manufacturing output bias during the study period. The $CO_2$ analysis results show that most of the technology level of China's manufacturing industry is still based on fossil fuels as the main energy demand. It is clear that the development and supply of green energy did not become the main demand of China's manufacturing industry during the study period. Industry scale has a negative impact on technological progress. This may be due to the emergence of corporate monopolies within the industry when the industry scale is oversized. As shown in the analysis in Section 4.2, China's manufacturing industry is more inclined to labor than capital and energy. The development of China's manufacturing industry, which is dominated by labor-intensive industries, is more dependent on China's demographic dividend. Technological progress dominated by green innovation often represents higher R&D investment. Obviously, Chinese manufacturing companies' monopoly has not been able to promote the continuous research and development of technology but has inhibited technological progress.

**Table 6.** Threshold estimates for China's manufacturing sector biased-technological changes.

| Variable | lncIBTC | | Variable | lncOBTC | | Variable | lncBTC | | Variable | lncTC | |
|---|---|---|---|---|---|---|---|---|---|---|---|
| | Coef. | t-Stat. | | Coef. | t-Stat. | | Coef. | t-Stat. | | Coef. | t-Stat. |
| lnER ≤ −3.822 | 0.003 *** | 1.250 | lnER ≤ −3.6218 | −0.02 *** | −4.600 | lnER ≤ −1.8297 | −0.006 ** | −2.22 | lnER ≤ −2.2132 | 0.052 *** | 6.270 |
| −3.822 < lnER ≤ −2.6491 | −0.004 | −1.030 | −3.6218 < lnER ≤ −2.6790 | 0.0002 | 0.030 | −1.8297 < lnER ≤ −1.2931 | −0.043 *** | −3.89 | lnER > −2.2132 | −0.021 | −0.960 |
| lnER > −2.6491 | −0.023 | −0.850 | lnER > −2.6790 | −0.021 *** | −2.470 | lnER > −1.2931 | −0.025 ** | −2.07 | - | - | - |
| lnPROP | −0.032 *** | −6.200 | lnPROP | 0.026 *** | 3.560 | lnPROP | −0.004 | −0.81 | lnPROP | −0.010 | −0.610 |
| lnR&D | 0.020 *** | 3.300 | lnR&D | −0.012 | −1.400 | lnR&D | 0.005 | 0.85 | lnR&D | 0.117 *** | 6.210 |
| lnEDV | 0.002 | 0.680 | lnEDV | −0.018 *** | −4.290 | lnEDV | −0.011 *** | −4.05 | lnEDV | 0.032 *** | 3.320 |
| lnASE | 0.038 *** | 7.910 | lnASE | −0.069 *** | −10.830 | lnASE | −0.03 *** | −7.40 | lnASE | −0.036 *** | −2.430 |
| lnSEC | 0.013 | 1.340 | lnSEC | −0.011 | −0.870 | lnSEC | 0.002 | 0.25 | lnSEC | 0.116 *** | 3.930 |
| Cons. | 0.009 | 0.25 | Cons. | 0.849 * | 1.48 | Cons. | 0.115 *** | 3.1 | Cons. | 0.811 *** | 6.26 |

Note: ***, **, and * denote statistical significance at the 1%, 5%, and 10% levels, respectively.

## 5. Conclusions and Policy Implications

Based on incorporating the SBM measurement model and the Malmquist productivity index, this paper measures the green total factor productivity, technological progress, and the input-biased and output-biased technological change of China's manufacturing industry from 2003 to 2015. It also investigated the mechanism of environmental regulation, property right structure, enterprise-scale, energy consumption structure, and other factors on technological progress and its decomposition items. Admittedly, we have noticed that this article may have some limitations. For example, we have not separately analyzed the appropriate environmental regulation policies for certain specific industries. Because we focus more on the perspective of technological progress. In the next research, we will start from the industry perspective and examine the relationship between pollution levels and the application of green technology to environmental regulations. This paper provides significant value and meaning as a powerful reference for adjusting policies to improve the green and sustainable development in China's manufacturing sector. According to the previous analysis, there are several conclusions and recommendations as follows:

(1) During the study period, the growth rate of green total factor productivity in China's manufacturing industry showed an overall downward trend, with an average annual growth rate of −0.622%. As for the decomposition, the average annual growth rate of technical efficiency EC was −0.105%. Technological progress TC is −0.497%. Compared with efficiency changes, technological changes are the main reason for the decline of green total factor productivity, which accounts for 79.9% of the decline in the growth rate of green total factor productivity. The average value of the BTC index is 1.003, which is very close to 1. This indicates that the overall technological progress of China's manufacturing industry during the sample period is a neutral technological change, but the technological progress bias between industries is more obvious with significant differences.

(2) By analyzing the elements of the Chinese manufacturing subsector combined with technical progress bias, we found that, in the input bias, the manufacturing sector showed significant L-Using/k-saving, L-using/E-saving, and K-using/E-saving are factor-biased characteristics. This shows that China's manufacturing sector was mainly labor-intensive industries during the study period, with obvious labor input preferences, which benefited from China's demographic dividend. In the output bias, the manufacturing industry has a clear bias toward the undesirable output $CO_2$ characteristics, which indicates that the sector still uses fossil energy as the main energy consumer. The input of fossil energy has positive significance in promoting the technological progress of the sector. This also shows that green energy has not yet occupied China's manufacturing industry's main energy consumption, and the level of green technology innovation needs to be improved.

(3) Through the analysis of the threshold model of environmental regulation on the technological progress of China's manufacturing industry. Environmental regulation has obvious dual-threshold characteristics for biased technological change bias (BTC), input-biased and output-biased technological change, as well as obvious single threshold characteristics for technological progress, which shows that the relationship between environmental regulation and technological progress is not a simple linear relationship. For BTC, environmental regulation has a negative impact on BTC. As the intensity of environmental regulation increases, its impact on BTC first increases and then decreases. For IBTC, as the intensity of environmental regulations increases, its impact on IBTC has changed from positive to negative. For OBTC, as the intensity of environmental regulations increases, its impact on OBTC has changed from negative to positive and then to negative ultimately. For TC, as the intensity of environmental regulations increases, its impact on TC has changed from a positive impact to a negative impact. Regulating the intensity of environmental regulations can improve the technological progress of China's manufacturing industry in a targeted manner. For example, for resource conservation requirements, more attention is paid to the

intensity of IBTC's environmental regulations; for pollution reduction requirements, more attention is paid to the intensity of OBTC's environmental regulations.

(4) In terms of other influencing factors, except for the property rights structure (PROP), the other influencing factors are all significant at the 1% level. Among them, R&D intensity, industry experts, and energy structure positively impact technological progress. The scale of the industry has a negative impact. This shows that R&D and foreign exports can effectively promote technological progress. At present, China's manufacturing industry still uses fossil energy as the primary energy consumption, and all factor inputs are more biased towards labor factors. This analysis result is also consistent with the previous conclusion on technological progress biased factors.

Based on these conclusions, this paper puts forward the following policy suggestions in the hope of providing policy enlightenment and help for China's greening process. First of all, through the analysis of the article, we have noticed that the expansion of China's manufacturing industry mainly relies on economies of scale. There is still a certain gap in technology and total factor productivity compared with developed countries. Only by promoting the development of innovation and environmental protection technology can we achieve sustainable economic development. Therefore, enterprises should be encouraged to make technological innovation and increase investment in green technology-related fields. In fact, the Chinese government has implemented policies targeting sectors, such as electric vehicle production subsidies and photovoltaic industry support and issued "Guidelines on building a market-oriented green Technology Innovation System" and other policies, leading the banking and financial institutions to finance green technology innovation enterprise and project effectively. It is a long process from research and development to innovation of environmental protection and energy-saving technologies, but only by relying on innovation and environmental protection technologies to realize the sustainable and healthy development of China's manufacturing industry can we avoid excessive dependence on resources and the environment.

Secondly, the strength and implementation scope of environmental regulation should be flexibly grasped. As the results of this paper show, from the perspective of environmental regulation strength, the most suitable policy intensity is different for different aspects of technological progress. Different from existing related studies that directly use environmental regulations in green TFP or technological progress, we decompose technological progress into biased technological changes and scale technological changes, and further decompose biased technological changes into input and output biased technological changes. The analysis of the impact of environmental regulations on these decomposing items will help to study the relationship with environmental regulations from the internal perspective of technological progress. For example, environmental regulations have different effects on the technological bias of input and output projects. From the perspective of pollution emission, the focus should be on the environmental regulation intensity of output-oriented technology progress bias. Further, from the perspective of resource-saving, the focus should be on the environmental regulation intensity Input-oriented technology progress bias. In response to different needs, with the perspective of circular economy and sustainable development [56], specifying a reasonable intensity of environmental regulation can achieve the purpose of reducing greenhouse gas emissions and resource conservation, and effectively promote the green technological progress of the manufacturing industry, which is beneficial to the realization of circular economy and sustainable development [57,58]. In addition, the formulation of environmental regulations can differentiate between companies that have applied or are developing green technologies and companies that have not invested in green technologies. Charge corresponding environmental protection fees to companies that do not have green R&D investment, which can promote the green transformation of some companies with backward production methods. Moreover, from the perspective of the scope of environmental regulation, it can effectively regulate the benign promotion between environmental protection and economic development by formulating corresponding environmental regulation policies for different industries. For example,

according to the degree of green technology application or pollution emissions in different industries, environmental regulations of different intensities are formulated to avoid one-size-fits-all policies.

Finally, in order to achieve the goal of both technological progress and green sustainable development. Enterprises are supposed to be encouraged to export, which will help the introduction of advanced technologies. The increase in international trade will help expand the scale of domestic industries, and enterprises can learn advanced international experience and related technologies through export behavior and gain trade spillovers. Increasing the value of exports enables enterprises to acquire green technologies and eliminate existing polluting technologies. It must be noted that the coefficient of the energy structure index is positive, which shows that the current energy consumption demand and technological system of China's manufacturing industry are still based on coal consumption. Therefore, the energy consumption structure should be changed and increase the proportion of green energy applications.

**Author Contributions:** Conceptualization, M.D. and W.L.; methodology, M.D; software, M.D.; validation, W.L., M.D. and Y.B; formal analysis, M.D.; investigation, M.D.; resources, M.D.; data curation, M.D.; writing-original draft preparation, M.D.; Y.B.; writing review and editing, M.D; visualization, M.D.; supervision, W.L.; project administration, W.L.; funding acquisition, W.L. All authors have read and agreed to the published version of the manuscript.

**Funding:** This study was financially supported by the Key Research Base project for Humanities and Social Sciences of the Ministry of Education of China (16JJD790015).

**Institutional Review Board Statement:** Not applicable.

**Informed Consent Statement:** Not applicable.

**Data Availability Statement:** The raw/processed data required to reproduce these findings cannot be shared at this time as the data also forms part of an ongoing study.

**Acknowledgments:** The authors are grateful to the editor and the anonymous reviewers of this paper.

**Conflicts of Interest:** The authors declare no conflict of interest.

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
