# Peer review of "Mechanisms of Environmental Regulation’s Impact on Green Technological Progress—Evidence from China’s Manufacturing Sector"

_sustainability, doi:10.3390/su13041600_

Round 1
Reviewer 1 Report
The answer on question "Does the introduction provide sufficient background and include all relevant references?" was marked as "not applicable" because "environmental regulation" is presented as problem of authors. For example sentence "... TFP of China's 42 manufacturing industry through reasonable environmental regulation and control" IS A WISH, A SENTENCE "goal is
to analyze the mechanism by which environmental regulations and other factors influence the greening of technological progress" HAVE LEFT TFP FOR GREENING, AND SENTENCE ASSOCIATES "... environmental regulations" WITH "and other factors on technological progress" AND "intensity of environmental regulations".
The answer on question "Is the research design appropriate" was marked as "not applicable" because abbreviations in methodology, suc as E NCV CEF arre not plroperly explained. I would recommend write its meaning and put the abbreviation to bracelets behind it. Other problem is that environmental regulation is based on "technological progress, Fare et al. ( 1997 " and "technological efficiency change". Environment is not mentioned. "inputs in M" are renamed to "s, y s, b s (sorry for changed format) representing the slack values of the input, good output, and bad output when S is desired outputs and I undesired outputs". "not applicable". Why I is output and M and s are inputs? Shouldn't be input (I) and output (O)?
Question "Are the conclusions supported by the results" is marked as "not applicable due to first sentence of conslusions "Malmquist productivity index method IN this paper measures the green total factor productivity". How index can measure total? How total productivity relates to title of the article "environmental regulation"? Is it possible to express the mechanism of environmental regulation by the "index"? How many years will China need to reuce CO2 emissions before 1990 year based on this index based method?
Sentences with recommendations in conclusions like "First, we should continue (WHY TO CONTINUE?) to encourage and induce technological innovation (WAS IT OBSERVED?)in enterprises, deepen investment and development (WAS IT OBSERVED?) of green technologies, and rely on innovation and environmental protection technologies (IS THERE ANY DIFFERENCE BETWEEN "PROTECTIVE" AND "GREEN" OBSERVED IN THE ARTICLE?) to achieve the sustainable and healthy (IS THERE ANY DIFFERENCE BETWEEN "SUSTAINABLE" AND "HEALTHY" OBSERVED IN THE ARTICLE?) development of China's manufacturing industry to avoid excessive dependence on resources (THIS "dependence on resources" HAVE APPEARED IN CONCLUSIONS THE FIRST TIME) and the environment. "
Typos in title "2.1 Related Research on the Bias of Technological Chang", double "are" in the sentence "there are several conclusions and recommendations are as follows:", dot "⑴ .During", font and dot in "⑶.Through", " " double spaces should be corrected.
Reviewer 2 Report
- The title of the article stresses the regulatory aspects of green technological change in the Chinese industry. This might be considered as putting the preasure for legal and managerial instruments. The article hovewer is purely economical and uses economic methods of analysis. Reviewer would suggest changing the title so that the field of research would be better represented in the title.
- The indstroductionary part seems to be a bit long. It could be shortened.
Reviewer 3 Report
The overall aim of the paper is to investigate the influence of environmental regulation on the greening of technological progress in China’s manufacturing industry. The paper analyses the influence of key input factors on technological progress and on greening total factor productivity in China’s manufacturing industry. In order to achieve this, the paper first reviews the literature. It presents the statistical model and the data utilised, to then summarize the research findings and discuss the policy implications in the final section.
Broad comments highlighting areas of strength and weakness.
The paper is well-presented with a solid overall structure. It presents an interesting topic with a relevant focus on the role of environmental regulation in transforming manufacturing industry.
With a surge of studies addressing TFP estimates in China’s industry using different quantitative methods, I would have expected the authors to provide a more thorough review of the literature linking TFP, green TFP and manufacturing in China. As it is, it is difficult to gauge the state of play and the paper does not adequately review the extant literature that deals with green TPF, environmental regulation and technology bias in the manufacturing industry in China. This is a significant shortcoming. For example, in a study on green TFP in China’s industrial economy, Chen et al., (2018) are moderately optimistic about the impact of environmental regulation on green TPF; whereas Li and Wu (2017) suggest reinforcing environmental regulation whilst taking into account regional specificities. How does the paper build on these and other similar studies (e.g. Shi and Li, 2019)? Following from the above, the research rationale and the focus on an industry-level analysis is not adequately justified.
Moreover, the reader does not get a clear sense of the structure of the manufacturing industry in China which is not described in the paper (e.g. Chen et al., 2018 distinguish between heavily-polluting and light polluting industries). Also the environmental regulatory regime governing the manufacturing sector and how this might be impacting greening is not discussed. As a result, the research gap does not come out very strongly; and the contributions are not sufficiently well identified.
Specific comments:
Abstract:
The abstract is well-written and clearly summarizes the research gap and contributions of the research.
Section 1 – Introduction:
The Introduction effectively describes the research aim. However, I feel that this section needs to be enriched and improved by providing a stronger rationale for selecting the manufacturing industry as the analytical focus, providing definitions of key terms and how these key terms are used in the context of the study. The authors make several assumptions in the introduction about total factor productivity growth and the factors influencing TFP at industry level; and it would be good to provide an overview of the manufacturing industry in China (size, main industry types etc.). There are too few references in this section. When quoting “previous research” in line 43 the authors are encouraged to specify which research is being referred to and who the key authors of these studies are.
Line 44: Insert one/more relevant references to previous research.
Line 76: specify which other influencing factors you refer to here
Section 2 – Literature Review:
The section does not make sufficient reference to relevant studies on TFP estimates in Chinese industry using different measuring methods – see my general comments above. It would be interesting to include a reflection on the evolution of environmental regulation in China in the period under study (2003-2015), highlighting whether environmental policy has served as an inflection point steering the direction of technological bias in manufacturing. This is alluded to in lines 347-351 in Section 4 and in Line 536 of the Conclusion (quote: “The green and sustainable development policy regulation”)
Line 88: typo "change" rather than "chang"
Line 122..."and [delete such as] Rubashkina (2015) tested the Porter hypothesis.....
Line 130: Author name(s) missing
Line 138: text is missing, "..........the impact of Chinese (?)
Section 3. Methods & Data
The authors could provide a justification for selecting the 2003-2015 time period for the analysis – is this related to the availability of data or does it reflect significant shifts in environmental regulation in China?
Section 4 – Empirical Results & Discussion
Generally, I feel that Section 4 somewhat lacks an eye for detail in emphasizing the differences and similarities across different sub-sectors of China’s manufacturing industry in terms of technology bias, greening of TFP and the impact of environmental regulations. For example, in Section 4.1, it is unclear how the bias of technological progress deviates from industry to industry (Lines 374-375). Also, the Section does not clearly indicate how the variability in technology bias differs between sub-sectors in the manufacturing industry (lines 377-379) – which are these sub-sectors? Can you specify or give examples from the tables provided?
Line 339: specify which “previous studies” your results are consistent with.
Line 357: which are those enterprises/sub-sectors that, quote: “face a crisis of obsolescence”
Lines 465-469: Rephrase the sentences since they lack clarity (text appears to be missing).
Line 471: which are the “related studies” you allude to in the text? (provide adequate references(s))
Line 507: The sentence, quote: “R&D intensity can increase the company” lacks clarity and needs to be rephrased.
Section 5 – Conclusions & Policy Implications
The main conclusions from the statistical analysis are summarized in Section 5.
The policy recommendations are relevant. However, these are rather general in scope and specific recommendations for designing and implementing policies that facilitate a systemic transformation of the manufacturing industry would be warranted. It would also have been interesting had the authors developed policy recommendations for specific sub-sectors within the manufacturing industry; and to reflect post-2015, on how the results obtained might be expected to evolve. It would be relevant to acknowledge the limitations of the statistical model used and provide directions for a future research agenda.
Line 582: it is not clear what the authors mean by, quote: “flexibly grasp the intensity and scope of environmental regulation”. Rephrase to clarify.
References
Additional references that could be relevant for this study:
Li and Wu (2017) Effects of local and civil environmental regulation on green total factor productivity in China: A spatial Durbin econometric analysis, Journal of Cleaner Production, 153, 342-353.
Shi and Li (2019) Green total factor productivity and its decomposition of Chinese manufacturing based on the MML index:2003-2015, Journal of Cleaner Production,222, 998-1008.
A further important issue:
Throughout the paper: CO2 - "2" to be written as a subscript. Check and correct throughout the manuscript.
Round 2
Reviewer 1 Report
Text contains repeatedly sentences "l. Error! Reference source not found." "Bias of technological progress" should be the main output. But, it is not properly defined. Definition of "bias of technological progress" is on row 626 as "internal influence mechanism of environmental regulations" Those internal regulations are specified as "internal impact mechanism's perspective of environmental regulation" at row 149. And, those perspectives are further specified as "internal relationship between environmental regulations and technological progress" at row 529.
Sentence "Drawing on the ideas of Weber and Domazlicky Error! Reference source not found. and Li et al. Error! Reference source not found. on the discriminative approach to the relationship between the direction of technological change and the elements" should conclude what of listed features was applied and how. Yes, I know that table 1 brings more details but reader can give up when reading such sentences.
Space should be inserted behind sentence ending dot, for example "reduction.However, in this period of technological change, some companies that have not developed and applied green technologies are facing a crisis of obsolescence, which is also the growth rate curve of MATC.The reason"
Citations "growth.[1-4] Neoclassical" and "is non-neutral[6-9], Kaneko [10], Zhao E" differ and should be unified.
There is no content in conclusion "(3). Through the analysis of the threshold model of environmental regulation on the technological progress of China's manufacturing industry, we have carefully analyzed the internal influence mechanism of environmental regulation on the technological progress of the sector. (THIS SENTENCE IS GOOD FOR INTRODUCTION.) Environmental regulation has obvious dual-threshold characteristics for biased technological change bias (BTC) and input-biased and output-biased technological change advancement, and obvious single threshold characteristics for technological progress itself, which shows that the relationship between environmental regulation and technological progress is not a simple linear relationship, the regulation of the intensity of environmental regulation can improve the technological progress of China's manufacturing in a targeted manner. (YES, CAN. SO WHAT?)" ADD HOW TO DO IT TO SIMPLIFY OPERATIONALISATION OF YOUR CONCLUSIONS BY MANAGERS, PLEASE. IOR REMOVE CONCLUSION 3.
SENTENCE " This shows that R&D and foreign exports can effectively promote technological progress". IT WAS KNOWN BEFORE THE RESEARCH HAS STARTED.
SENTENCE "Based on these conclusions, this paper puts forward the following policy suggestions in the hope of providing policy enlightenment and help for China's greening process" IS DELEGATING POWER OF CONCLUSIONS OF THIS ARTICLE TO POLITICIANS. WHY? CONCLUSIONS OF THIS ARTICLE SHOULD BE OPERATIONALISED FOR ERWS OF COMPANIES TO HELP THEM TO DEVELOP A COMPETITIVE ADVANTAGE. THEREFORE, WORD "ENCOURAGE" IN SENTENCE " First of all, continue to encourage and induce enterprise's technological innovation, deepen investment and development of green technology" SHOULD BE REMOVED AND REPLACED BY FIRM RESULT PROVING EFFECTS OF "investment and development of green technology".
IT SHOULD BE MADE CLEAR WHETHER THIS ARTICLE " examines the internal influence mechanism of environmental regulations" OR "Chinese government".
AFTER ROW 661 AND "(4)." CONCLUSION FOLLOWS AT ROW 679 "Secondly, the strength ..."
Conclusions do not interpret reached results properly. CAPITAL CHARACTERS WERE USED TO DISTINGUISH TEXT OF AUTHORS FROM TEXT OF OPPONENT.
Reviewer 3 Report
Introduction:
Delete Lines 29-31: the sentence seems irrelevant.
Lines 34-37: insert (a) reference(s) for the values/data quoted.
Line 49-51. Rephrase the sentence by stating that: “Environmental regulation acts as a driver to realize China's green development and has been shown to promote green TFP of China's manufacturing industry”.
Line 57: insert a description/definition of “technological progress bias” as used in the research.
Line 67-70 “The way to achieve this purpose is to analyze the influence mechanism of environmental 68 regulations and other factors on technological progress toward greening, so as to achieve the 69 dual goals of environmental protection and healthy economic development.” Place this sentence after Line 51 and rephrase to: “This paper analyses the influence mechanism of environmental regulations on technological progress toward greening, so as to achieve the dual goals of environmental protection and healthy economic development.” You also need to elaborate what you mean by “influence mechanism” of environmental regulation by deconstructing and teasing out what you mean by this – although it is obvious to you, it is not necessarily clear to the reader.
Section 2 – Literature Review:
Line 120:” meanwhile testing the "Porter hypothesis" and so on.”. Delete “and so on”.
Line 148-149. You need to explain what you mean by “internal impact mechanism’s perspective of environmental regulation”. Although this may be clear to the authors, it is not necessarily clear and obvious to the reader and it is a key sentence because it explains the gap that this research is addressing. Actually I make a similar comment above in the “introduction” section. This is very important.
Lines 198-210: fit better in Section 2.1 on “Related Research on the Environmental regulation and Green total factor productivity”
Section 3. Methods & Data
Line 329-331: the justification for selecting the 2003-2015 time period for the analysis is still not convincing. Why is data from this period more “complete and reliable” compared to other time frames? Is this related to changes in data collection approaches? New/other sub-sectors included? elaborate
Section 4 – Empirical Results & Discussion
In response to my comment about the need to highlight differences and similarities across different sub-sectors of China’s manufacturing industry in terms of technology bias, greening of TFP and the impact of environmental regulations, the authors comment with the following text which I think they could integrate in the paper in Section 4 as it provides a useful discussion on how the results are being analysed:
“Based on the relevant literature, there are not many articles about the details of the technological progress of China's manufacturing industry [rephrase this sentence and provide adequate references, recalling what was discussed in Section 2]. The interpretation of this part the results can effectively enable us to better understand the current situation of China's manufacturing industry and pave the way for the following environmental regulations.
As shown in Figure [refer to figure number], the trends and sizes of the maximum and minimum values are very different. This shows that with time changes, the specific details of technological progress fluctuate greatly. Our goal is not to emphasize the development of a specific industry, but to explore the relationship between such fluctuations and environmental regulations based on analysis. Of course, the analysis of R&D, exports, property rights structure, and energy consumption structure is also very important.”
Section 5 – Conclusions & Policy Implications
Line 639: avoid the use of absolute terms such as:” the differences are enormous.”. Specify the extent of the different and then use terms such as ‘significant’.
Lines 631-651: it seems to me that these results are not new and have been described by previous research. It would be good to acknowledge when your results build on previous by stating this in the text and providing relevant references.
In the conclusions and the recommendations, I think you need to bring back the notion you mention in the literature review about the “internal impact mechanism’s perspective of environmental regulation” and unpack this phrase. You do explain this in Lines 679-692. However, you need to be more explicit in indicating that the “internal mechanisms” you refer to relate to the impact of environmental regulation on different aspects of technological progress that you describe in lines 679-692.
Lines 669-678: The text included in Lines 669 to 678 is relevant though it is not clear how this emerges from the outcomes of the statistical analysis. For example, it is indeed important to encourage enterprise’s technological innovation, “deepen investment and development of green technology” for those industry sub-sectors that invest in R&D. What about the sub-sectors that rely on technology developed elsewhere? The Chinese government is implementing a number of “environmental policies” (“such as electric vehicle production subsidies and photovoltaic industry support, and issued " Guidelines on building a market-oriented green Technology Innovation System” and other policies” which are likely to reflect the results obtained .
Lines 675-678: I am not sure how the following sentence ties in with the outcomes from the empirical analysis in this paper (“It is a long process from research and development to innovation of environmental protection and energy-saving technologies, but only by relying on innovation and environmental protection technologies to realize the sustainable and healthy development of China's manufacturing industry can we avoid excessive dependence on resources and the environment”). Does manufacturing in China rely primarily on environmental protection technologies for sustainability? What is the role of end-of-pipe technology? The role of innovation (and investments in R&D) is not an aspect that is particularly well-developed in this paper. Naturally so, because this is not the focus of the analysis. Perhaps therefore that the sentence is rather too broad, and makes too many assumptions which are clearly not derived from your own research – best delete!
Lines 690-692: the claim that there is no one size fits all for environmental policy and that environmental policy needs to be tailored to different industries is interesting though you do not substantiate this claim based on the results of your analysis. Again, what type of environmental policies would be tailored to manufacturing? This is still vaguely addressed.
I think the Conclusion needs to be enriched further with recommendations for managers. For example, you distinguish between those companies that have invested in R&D or are applying green technologies and those companies that have not invested in green technologies. Which course of action would managers in these different categories/types of firms adopt based on the impact that environmental regulation is likely to have on green TFP in these companies?
I think that the conclusion needs much more work in terms of developing a critical analysis of the research outputs and their implications for policy and practice. As it is, the policy implications are still far too general in scope and there are no recommendations on the implications for managers in manufacturing companies. Moreover, the authors have not discussed any limitations of their statistical model and there are no explicit indications of how they would take this work forward.
Round 3
Reviewer 1 Report
The answer for sentence "Is the content succinctly described and contextualized with respect to previous and present theoretical background and empirical research (if applicable) on the topic?" is marked "Must be improved" because fulfilled "goals of environmental protection and healthy economic development" are not mentioned neither in conclusions nor in abstract. Sentence in conclusions "Finally, in order to achieve the goal of both technological progress and green sustainable development" is only repeated goal, not result. Sentence of abstract "The impact of environmental regulations on the Chinese manufacturing industry's technological progress has a significant threshold effect." doesn't inform whether the threashold is the result of this article or future mission for industry. Wishes as "Enterprises should be encouraged to export, which will help the introduction of advanced technologies, enabling enterprises to adopt green technologies and eliminate existing backward technologies" should reffer to data. This result delegates responsibility to industry by plenty of words "should".
For question "Are the research design, questions, hypotheses and methods clearly stated?" the answer is marked "must be improved because "A slacks-based measure (SBM)" is called " SBM undesired output model". Why? The sentence "Therefore, this paper adopts the SBM model with undesired outputs proposed by Tone, [55] which can consider the relationship between inputs, outputs, and undesirable outputs, to solve the slack problem in efficiency evaluation better" shouldn't contain word "can". All three values should appear in conclusions and abstract instead of threshold. It is not clear whether the outpout is undesired like in sentence "the SBM model with undesired output" or undistinguishable like in sentence "with the SBM indistinguishable undesired output model".
Answer to question "For empirical research, are the results clearly presented?" is marked as "must be improved because the sentence "This shows that China's manufacturing sector was mainly labor-intensive industries during the study period, with obvious labor input preferences, which also benefited from China's demographic dividend", which follows after the sentence "In the output bias, the manufacturing industry has a clear bias toward the undesirable output CO2 characteristics." Otherwise you should prove how much CO2 produces labour intensive production. I would remove the sentence and leave only the sentence about CO2 related to fossil energy. Results of BTC on rows 681 - 690 do not justify threshold as the main output information of this article in its abstract. Paragraphs 2-4 contain no values. Conclusion is too long. I would remove all sentences delegating work to someone else like "First of all, continue to encourage and induce enterprise's technological innovation, deepen investment and development of green technology". The sentence may remain in the text if quantified and proven benefits will be added.
Sentence of conclusions "We also analyzes ... " needs correction of English.
Reviewer 3 Report
Thank you for addressing my comments in the second round. The paper has improved considerably though I think that before being published, the paper needs to be thoroughly proof read by a native English speaker, also to correct several typos and grammar mistakes.
Below are some additional minor corrections that need to be made. Good luck with your research!
Line 144-147: either delete or rephrase as not clear enough
Line 196 cao in capital letters, therefore “Cao”
Lines 295-300: sentences are repeated therefore delete the following sentences: “Reflects the changes in the marginal substitution rate of various input factors caused by technological progress. The output-biased technological change indicates the degree of technological change of different outputs when the input remains unchanged. Reflect the promotion of technological progress on multiple outputs.”
Lines 647-654: the limitations and suggestions for future research should be shifted further down e.g. insert just before line 699.
Line 736: it is not clear why, quote: “Enterprises should be encouraged to export” in order tointroduce advanced green technologies. Eliminate first part of sentence. Also instead o f the term “backward” technologies in line 738, insert “outdated or polluting technologies”.
